# DIAL-G²: GRAPH-GUIDED DIALECTICAL AGENT FOR ADVANCED ESG REASONING

## ABSTRACT

The rapid growth of the focus on Environmental, Social, and Governance (ESG) creates the need for an effective AI tool for evaluating corporate sustainability. Despite the increasing availability of ESG data disclosed by companies, the complex, relational, and unstructured nature of ESG reports poses significant challenges for large-scale LLMs. The surprising efficacy of small language models (SLMs) on domain-specific tasks challenges the prevailing belief that only massive, general-purpose models can tackle complex reasoning. Inspired by this observation, we investigate the potential of compact models in the nuanced field of ESG analysis, arguing that the path to true expertise lies in focused knowledge and structured collaboration. Our work presents a two-fold contribution to realize this vision: 1) We introduce ESGEXPERT-30K, a new knowledge-intensive dataset, to fine-tune a compact SLM into a specialized ESGEEK model that achieves state-of-the-art accuracy on domain-specific QA. 2) We propose DIAL-G², a novel framework where these expert agents form a committee to analyze full, multimodal corporate reports. In DIAL-G², agents populate a shared graph with conflicting arguments. The GNN then performs relational inference on this graph and uses its learned attention weights to direct the agents' focus towards the most salient and contested information in a subsequent debate phase. This graph-guided approach demonstrates remarkable effectiveness. Furthermore, on our newly contributed, large-scale ESGREPORT-RATING-50K benchmark, DIAL-G² achieves human-expert-level performance in end-to-end ESG score prediction. Extensive experiments show that DIAL-G² can overcome bottlenecks of Multi-Agent Systems (MAS) in domain-specific understanding, providing the ESG research community with a powerful new paradigm for scalable, relational, and interpretable AI.

## 1 INTRODUCTION

Numerous studies have established a positive correlation between Environmental, Social, and Governance (ESG) performance and firm value (Gillan et al., 2021; Wong et al., 2021; Serafeim & Yoon, 2023). And stakeholders evaluate the ESG performance of companies based on their annual ESG reports. Unlike data from financial statements, regulators do not mandate ESG data filing in specific forms in most cases. Efforts to standardize ESG reports face challenges like complex regulations, evolving frameworks, and compliance verification (Mishra et al., 2024). With the rapid development of Large Language Models (LLMs), the call for evaluating ESG data using LLMs prevails.

However, research on LLMs for ESG has bottlenecks due to inherent framework-related limitations. While fine-tuned LLMs are applied to such tasks (Birti et al., 2025), knowledge shortage and hallucination make it not applicable. To ground LLMs in external knowledge and mitigate hallucination, Retrieval-Augmented Generation (RAG) variants, like multi-modal RAG (Wang et al., 2025b), hybrid RAG (Wang et al., 2025a), and graph-based RAG (Edge et al., 2025; Guo et al., 2025; Barghi, 2025), have become a dominant paradigm. However, these RAG-based approaches often follow a linear "retrieve-then-synthesize" workflow. They lack a mechanism for deep relational reasoning, iterative refinement, or for resolving subtle, conflicting evidence distributed across a lengthy document. Reinforcement Learning(RL) with LLM for ESG is also researched (Hou et al., 2025), but the utilization of general LLMs in the framework results in the lack of domain knowledge and massive resource consumption, making it not suitable for production scenarios.

In this paper, we propose a new paradigm that embraces specialization and structured collaboration. By fine-tuning on our knowledge-intensive dataset ESGEXPERT-30K, we have developed a series of efficient, compact language models with a deep understanding of ESG, which we refer to as ES-GEEKS. This ensures that our AI agents have the cognitive foundation to become experts in the ESG field. These expert agents form the core of our innovative framework, **Dial**ectical Agents with **G**raph-**G**uide (DIAL-G²). [1] It represents reports as an evidence graph where a GNN acts as a dynamic conductor, using attention to identify salient "hotspots" and guide a dialectical debate among ESGEEK-based agents. The outcome from the debate iteratively refines the graph's beliefs, creating a neuro-symbolic loop that enables deep, relational, and interpretable analysis beyond standard retrieval methods. We validate our approach through extensive experiments. We first confirm the SOTA performance of our ESGEEK model. Then, using our newly created ESGREPORT-RATING-50K benchmark, we show that the full DIAL-G² framework can predict a company's ESG score with a level of accuracy comparable to human experts. Our contributions are:

- We introduce and validate a method for creating highly specialized, efficient **ESGEEK** models by fine-tuning on our new, knowledge-rich **ESGEXPERT-30K** dataset.

- We propose **DIAL-G²**, a novel framework that deeply fuses a GNN's relational reasoning with the linguistic intelligence of specialist agents through a unique, iterative, graph-guided debate mechanism.

- We contribute **ESGREPORT-RATING-50K**, a large-scale, multimodal dataset for ESG score prediction, addressing a key resource gap.

- We demonstrate that our complete system achieves SOTA performance and provides a collaborative architecture paradigm composed of specialized small models for accomplishing complex real-world tasks.

## 2 RELATED WORKS

### 2.1 THE IMPORTANCE AND CHALLENGES OF ESG EVALUATION

Research has indicated a direct, positive impact of high ESG scores on a company's operational performance and efficiency (Chen & Xie, 2022). Furthermore, ESG information plays a crucial role in shaping market perception and reducing information asymmetry (Cornell, 2021). ESG-related controversies are shown to heighten perceived uncertainty surrounding a firm's future cash flows Schiemann & Tietmeyer (2022), while transparent and comprehensive ESG disclosures tend to mitigate this uncertainty by providing stakeholders with decision-useful information (Serafeim & Yoon, 2023). However, some studies suggest that in cases of manual rating, increased ESG disclosure may exacerbate rather than resolve ESG rating discrepancies (Christensen et al., 2022; Dimson et al., 2020), which contradicts our expectations. Consequently, leveraging artificial intelligence to address these pain points—enhancing objectivity, scalability, and the ability to process vast, unstructured data—has become a critical area of research, motivating the work presented in this paper.

### 2.2 DOMAIN-SPECIFIC MODEL SPECIALIZATION

The concept of creating expert models for high-stakes domains is well-established. Models like Fin-BERT (Liu et al., 2020) and BloombergGPT (Wu et al., 2023a) demonstrated the value of domain-specific pre-training. In medicine, models like Med-PaLM 2 have shown expert-level performance by fine-tuning on medical knowledge (Qian et al., 2024). Amila Silva et al. simultaneously retain domain-specific and cross-domain knowledge in multimodal data to detect fake news from different domains (Silva et al., 2021). As for the ESG field, Tim Nugent et al. combined domain-specific language models and data augmentation methods to detect ESG issues (Nugent et al., 2021), improving the accuracy of classification tasks. ClimateQA (Luccioni et al., 2020) was developed to classify whether a sentence from an ESG report answers regulatory questions. ESGenius (He et al., 2025) has established the first LLM evaluation benchmark focused on ESG and sustainable devel-

---

[1]Code, dataset, and appendices with full experimental details will be made publicly available at `https://anonymous.4open.science/r/dial_g2_ESGeek_framework-E613/README.md` for reproducibility.

opment knowledge QA. Our work, built on knowledge distillation and specialization for ESG QA, follows this lineage.

### 2.3 MULTI-AGENT SYSTEMS AND GRAPH-GUIDED REASONING

To move beyond the limitations of simple RAG, researchers are exploring more sophisticated reasoning frameworks. LLM-powered multi-agent systems (MAS) enhance the capabilities of a single model by orchestrating collaborative dialogues between specialized, role-playing agents (Chen et al., 2023a;b; Zhang et al., 2024a; Han et al., 2025). Concurrently, Graph Neural Networks (GNNs) (Velic̆kovic et al., 2018; Chen et al., 2018; Kipf & Welling, 2016) have become the standard for modeling relational data, proving highly effective in document analysis for tasks like modeling citations or logical flow (Liu et al., 2021; Wu et al., 2023b).

## 3 PROBLEM FORMULATION

Our research addresses two complementary challenges in automated ESG analysis, moving from granular, evidence-based inquiry to holistic, predictive assessment.

### 3.1 TASK 1: RELATIONAL ESG QUESTION ANSWERING (QA)

Given a corporate document $D$ and a specific ESG-related question $q$, our first task is to find a factually accurate and contextually complete answer. The question $q$ is assumed to be mappable to a primary topic node $v_q$ within a pre-defined ESG Knowledge Graph $G = (V, E)$. Unlike standard extractive QA, the desired output is a tuple $(a, \mathcal{C}, J)$, where $a$ is a synthesized answer, $\mathcal{C}$ is a set of supporting evidence quotes from $D$, and $J$ is a justification narrative. This task tests the framework's ability to perform deep, evidence-grounded reasoning.

### 3.2 TASK 2: MULTIMODAL ESG SCORE PREDICTION

This more challenging task aims to predict a company's overall ESG score directly from its full, multimodal annual report. Given a multimodal document $D_{multi}$, which comprises text $D_{\text{text}}$, a set of tables $\mathcal{T}$, and a set of images $\mathcal{I}$, the task is to learn a function $f : D_{multi} \to \mathbb{R}$ that predicts a single, continuous ESG score $\hat{y}$. The objective is to minimize a regression loss function, such as the Mean Squared Error (MSE), between the predicted score $\hat{y}$ and the ground-truth score $y$:

$$\mathcal{L}_{\text{score}} = (\hat{y} - y)^2$$

This task evaluates the framework's ultimate capability to perform holistic, multi-modal synthesis and predictive judgment.

## 4 METHODOLOGY

We introduce a framework for structured, collective intelligence, designed to surpass monolithic models in complex reasoning. As shown in Figure 1, it is based on three pillars: (1) The distillation of vast domain knowledge into a compact expert model; (2) An innovative, graph-guided dialectical architecture for orchestrating expert models; and (3) A robust mechanism for synthesizing their findings into a final predictive judgment.

### 4.1 THE ESGEEK MODEL: CREATING AN ESG EXPERT

The foundation of our framework is the specialized ESGEEK model, created through a two-step knowledge distillation process. First, to codify domain expertise, we built ESGEXPERT-30K, a new high-quality dataset with over 30,000 Q&A pairs. This dataset was meticulously crafted by using an LLM to extract knowledge from 12 seminal ESG standards, followed by multi-stage human and expert validation (see Appendix B for full details). Subsequently, we fine-tuned a pre-trained SLM (Qwen-2.5 0.5B) on this rich dataset, a process that distills comprehensive ESG knowledge into our final expert model, ESGEEK ($M_{ESG}$), a highly capable and efficient reasoning module.

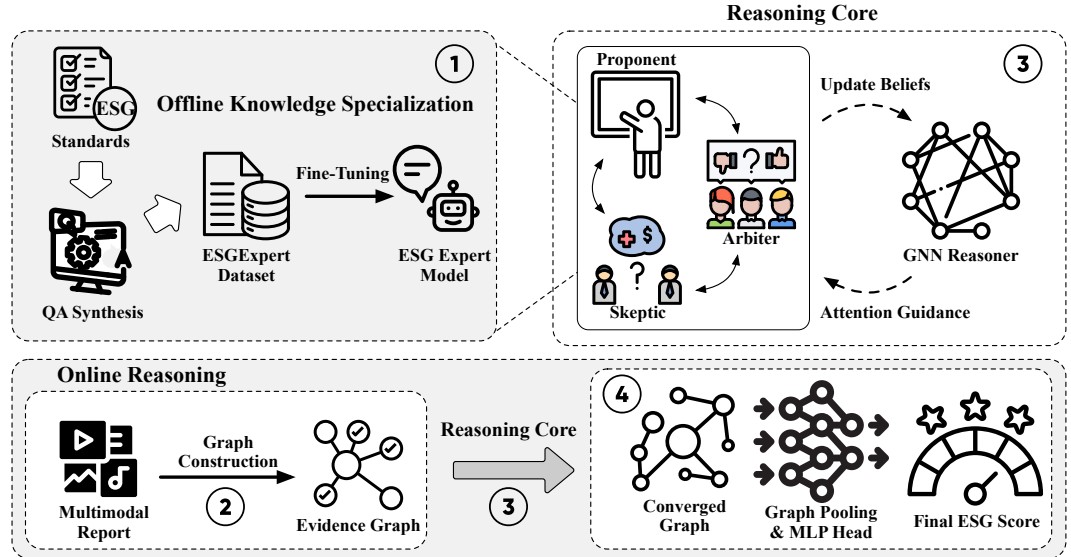

Figure 1: An overview of the DIAL-G² framework. The process consists of two components: (1) **Offline Knowledge Specialization**: An expert SLM (ESGEEK) is created by fine-tuning on our knowledge-intensive ESGEXPERT-30K dataset. (2) **Online Reasoning**: For a given multimodal report, an evidence graph is constructed. A GNN Reasoner then guides a "Reasoning Core", i.e., comprising a Proponent, a Skeptic, and a synthesizing Arbiter, to debate and iteratively update beliefs on the graph. The GNN's attention guidance directs the agents' focus to the most critical information. (3) **Final Prediction**: The converged graph representation is passed through a graph pooling and MLP head to generate the final ESG score.

## 4.2 THE DIAL-G² ARCHITECTURE

DIAL-G² orchestrates instances of $M_{ESG}$ within a graph-based reasoning framework.

### 4.2.1 EVIDENCE GRAPH CONSTRUCTION.

A report $D_{multi}$ is transformed into a graph $G = (V, E)$, where each node $v_i \in V$ represents a page. The edge set $E$ is crucial for defining the scope of relational reasoning. We construct a hybrid graph where an edge $(v_i, v_j)$ is created if any of the following conditions are met: (1) **Structural Adjacency**: Pages $i$ and $j$ are sequential. (2) **Hierarchical Links**: Both pages belong to the same section as defined by the document's table of contents. (3) **Semantic Similarity**: The cosine similarity of their page-level text embeddings exceeds a threshold of $\tau = 0.8$. This hybrid policy captures both narrative flow and long-distance thematic connections.

### 4.2.2 MULTIMODAL FEATURE ENCODING.

The initial feature vector $\mathbf{x}_i \in \mathbb{R}^{384}$ for each page-node is derived from its multimodal content. For textual content $T_i$, we use a pre-trained sentence-transformer, 'all-MiniLM-L6-v2', to generate an embedding $\mathcal{E}_{\text{text}}(T_i)$. For tables $\mathcal{T}_i$ and images $\mathcal{I}_i$, we first use 'Florence-2-large' to generate descriptive captions, which are then encoded by the same text encoder to form $\mathcal{E}_{\text{modal}}(\mathcal{T}_i, \mathcal{I}_i)$. All synthesized beliefs from the debate are also encoded using $\mathcal{E}_{\text{text}}$. The final feature vector is a learnable fusion via a two-layer MLP with GELU activation:

$$\mathbf{z}_i = [\mathcal{E}_{\text{text}}(T_i) \,\|\, \mathcal{E}_{\text{modal}}(\mathcal{T}_i, \mathcal{I}_i)]$$

$$\mathbf{x}_i = \text{MLP}_{\text{fuse}}(\mathbf{z}_i)$$

To illustrate this process, Figure 2 shows a sample report page and its parsed representation.

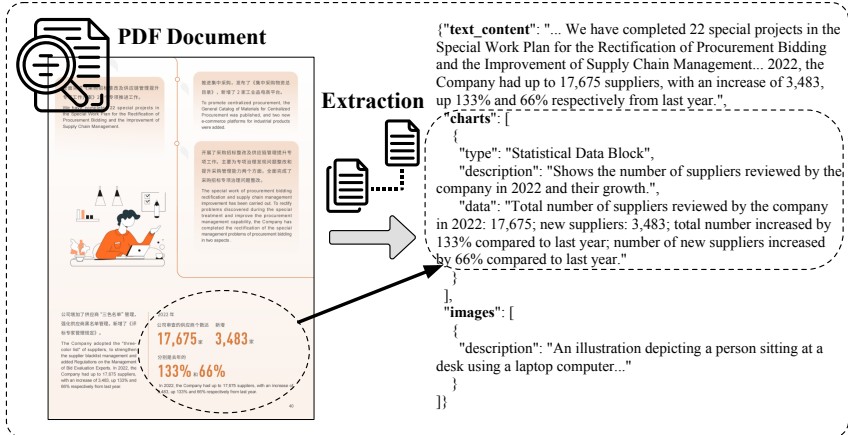

Figure 2: An example of multimodal content processing for a single document page. The raw page contains a mix of text, statistical data, and illustrative graphics. Our system parses this into a structured format, extracting the textual content, identifying and interpreting the chart's data (e.g., 17,675 suppliers), and generating a description for the image. This rich, structured information serves as the initial feature representation for a node in the evidence graph.

### 4.2.3 THE GRAPH-GUIDED DEBATE (GGD) WORKFLOW.

This neuro-symbolic loop refines a shared understanding of the document. The state of debate for node $v_i$ at iteration $k$ is stored in an Evidence Log, $\mathcal{E}_{v_i}^{(k)}$. Full prompt templates are provided in Appendix C.

**Belief Initialization ($k = 0$):** The Proponent ($A_{pro}$) and Skeptic ($A_{skep}$) agents generate initial conflicting beliefs ($s_{pro,v_i}^{(0)}$, $s_{skep,v_i}^{(0)}$) for each node $v_i$. The initial hidden state $\mathbf{h}_i^{(0)}$ is a learnable fusion of these beliefs.

**Iterative Refinement Loop (for $k = 0, \ldots, K-1$):** The loop consists of three main steps: GNN-driven discovery, dialectical exchange, and arbitrated synthesis.

1. *Attentive Relational Inference:* We use a GAT with Dropout and DropEdge (Zhang et al., 2024b) for robust training. It processes the graph state $H^{(k)}$ to produce updated intermediate states $\mathbf{H}'^{(k)}$ and attention weights $\{\alpha_{ij}^{(k)}\}$, identifying relational hotspots, that is, top-K attention-weighted edges.

2. *Dialectical Exchange on Hotspots:* For each hotspot edge $(v_i, v_j)$, the Proponent and Skeptic are prompted to generate arguments ($s'_{pro}, s'_{skep}$) that either support or challenge the connection between the content on these pages.

3. *Arbitrated Synthesis and Inquiry:* This is the core of the reasoning process. The Arbiter agent ($A_{arb}$) receives the arguments from both sides. Its task is two-fold:
   - **Synthesize Belief**: It generates a synthesized narrative $s_{\text{synth}}$, which integrates the valid points from both agents, highlights unresolved conflicts, and forms a nuanced judgment.
   - **Generate Inquiry**: Optionally, it can generate a targeted question $q_{\text{follow-up}}$ to probe the weaker argument, preparing for a potential deeper debate in the next iteration.

   The embedding of the synthesized belief, $\mathcal{E}(s_{\text{synth}})$, becomes the new linguistic insight for the node, denoted as $\mathbf{h}_{\text{rebuttal}}$.

4. *Gated Belief Update:* A gated mechanism fuses the GNN's propagated belief $\mathbf{h}'^{(k)}_i$ with the new linguistic evidence $\mathbf{h}_{\text{rebuttal},i}^{(k)}$ from the Arbiter's synthesis, ensuring a stable, iterative refinement of the node states to $\mathbf{h}_i^{(k+1)}$. The update is formalized as:

$$g_i^{(k)} = \sigma(W_g[\mathbf{h}'^{(k)}_i \| \mathbf{h}_{\text{rebuttal},i}^{(k)}] + b_g)$$

$$\mathbf{h}_i^{(k+1)} = (1 - g_i^{(k)}) \odot \mathbf{h'}_i^{(k)} + g_i^{(k)} \odot \text{MLP}_{map}(\mathbf{h}_{\text{rebuttal},i}^{(k)})$$

where $g_i^{(k)}$ is the gate vector, $\sigma$ is the sigmoid function, and $\text{MLP}_{map}$ is a mapping layer. This allows the model to learn how much of the new insight should influence the existing belief.

## 4.3 HOLISTIC JUDGMENT VIA GRAPH-LEVEL SYNTHESIS AND PREDICTION

The GGD workflow culminates in a set of final, converged node embeddings $H^{(K)} = \{\mathbf{h}_{v_i}^{(K)}\}_{i=1}^{N}$. Each vector $\mathbf{h}_{v_i}^{(K)}$ is no longer a representation of an isolated page, but a rich, context-aware belief state, refined by both linguistic debate and structural message passing. The final challenge is to synthesize these distributed, node-level beliefs into a single, coherent, and accurate judgment—the ESG score. This process unfolds in two steps: graph-level representation and score regression.

A simple aggregation mechanism, such as global mean pooling ($\mathbf{h}_G = \frac{1}{N} \sum_i \mathbf{h}_{v_i}^{(K)}$), would implicitly assume that every page of the report is equally important to the final ESG score. This is a flawed premise, as corporate reports invariably contain sections of varying materiality.

To address this, we employ a more sophisticated, attention-based graph pooling mechanism (Velic̆kovic et al., 2018). This allows the model to learn a materiality weighting over all nodes (pages) in the document graph, deciding which parts of the report are most indicative of the company's true ESG performance. The final graph-level representation $\mathbf{h}_G$ is computed as a weighted sum of the node embeddings:

$$\beta_i = \text{softmax}_i(\text{MLP}_{\text{pool}}(\mathbf{h}_{v_i}^{(K)}))$$

$$\mathbf{h}_G = \sum_{i=1}^{N} \beta_i \cdot \mathbf{h}_{v_i}^{(K)}$$

where $\text{MLP}_{\text{pool}}$ is a small, learnable network that outputs a single scalar attention score for each node. The softmax function normalizes these scores into a probability distribution $\{\beta_i\}$, where $\beta_i$ can be interpreted as the learned materiality or importance of page $i$ to the final prediction. This mechanism offers a critical layer of interpretability; by inspecting the nodes with the highest $\beta_i$ values, we can understand which parts of the report most influenced the model's final judgment.

The holistic graph embedding $\mathbf{h}_G$ encapsulates the distilled wisdom of the entire report. This vector is then passed through a final prediction head, $\text{MLP}_{\text{score}}$, to regress the continuous ESG score $\hat{y}$. This head consists of two feed-forward layers with a ReLU activation:

$$\hat{y} = \text{MLP}_{\text{score}}(\mathbf{h}_G)$$

It is crucial to understand the training dynamics. While the LLM-driven debate is a **non-differentiable** process, it is tightly coupled with the end-to-end trainable graph pipeline. The final prediction loss, $\mathcal{L}_{\text{score}} = (\hat{y} - y)^2$, backpropagates through the entire graph pipeline, creating an **indirect learning signal**. The GNN learns to compute attention weights ($\alpha_{ij}$) that are most effective at selecting "hotspot" edges, which, when presented to the non-differentiable debate module, are likely to produce synthesized beliefs that maximally reduce the final prediction error. This creates a self-optimizing system where relational reasoning and linguistic analysis synergize towards a single objective, without requiring gradients to flow through the LLM itself. The entire training process integrates these components into an end-to-end pipeline. The data flow for a single report involves: **(1) Initialization**, where the report is converted into a graph $G$ with initial beliefs $H^{(0)}$; **(2) Iterative Refinement**, where for $K$ steps, a GNN identifies salient edges to guide a dialectical debate whose synthesized outcomes update the node states $H^{(k+1)}$ via a gated mechanism; and **(3) Prediction**, where the final states $H^{(K)}$ are pooled to predict the score $\hat{y}$. The model's differentiable parameters are optimized by backpropagating the regression loss $\mathcal{L}_{\text{score}} = (\hat{y} - y)^2$. The complete procedural breakdown is shown as algorithms process in the Appendix.

## 5 EXPERIMENTS

We conduct a three-stage experimental evaluation to validate our claims. Full implementation details, model configurations, and hyperparameters for all experiments are available in Appendix E.

Table 1: QA performance on the ESGEXPERT-30K test set. Our fine-tuned ESGEEK models significantly outperform their base models.

| Model | Params | Accuracy (%) |
|---|---|---|
| Qwen-2.5-0.5B (Qwen et al., 2025) | 0.5B | 54.58 |
| **ESGEEK-0.5B (Ours)** | 0.5B | **76.42** |
| Qwen-2.5-1.5B | 1.5B | 63.91 |
| DeepSeek-R1-Distill-Qwen-1.5B | | 31.34 |
| **ESGEEK-1.5B (Ours)** | 1.5B | **71.21** |
| Qwen-2.5-3B | 3B | 58.89 |
| **ESGEEK-3B (Ours)** | 3B | **74.52** |
| Qwen-2.5-7B | 7B | 64.96 |
| DeepSeek-R1-Distill-Qwen-7B | | 50.18 |
| **ESGEEK-7B (Ours)** | 7B | **81.78** |
| GPT-4o-mini (OpenAI et al., 2024a) | 8B | 62.68 |
| Qwen-2.5-14B | 14B | 61.52 |
| Qwen2.5-Max | 325B | 64.44 |
| DeepSeek-R1 (DeepSeek-AI et al., 2025) | 671B | 66.29 |
| GPT-4o | 200B | 63.64 |
| o3 (OpenAI et al., 2024b) | 300B | 72.54 |

## 5.1 EXPERIMENT 1: VALIDATING THE ESGEEK

We benchmark the stand-alone QA capability of our domain-specialized ESGEEK on the held-out ESGEXPERT-30K test split. The task is formulated as multiple-choice question answering, where "Accuracy (%)" is the percentage of correctly answered questions.

Table 1 shows that our 0.5B specialist already surpasses GPT-4o-mini (8B) by 13.74 pp, while the 7B variant attains 81.78%—a +9.2 pp gain over the 300B o3 flagship and a +16.8 pp jump relative to its own base checkpoint, demonstrating that rigorous domain adaptation can distill expert competence into highly parameter-efficient SLMs.

## 5.2 EXPERIMENT 2: END-TO-END SCORE PREDICTION

We evaluate the full DIAL-G² framework on our ESGREPORT-RATING-50K dataset. This large-scale dataset, containing over 50,000 report-rating pairs, is designed for diversity and robustness. Labels were harmonized across multiple rating agencies by first Z-score normalizing ratings within each agency, and then mapping them to a unified 0-100 scale. Unlike benchmarks that focus solely on large-cap firms, our dataset includes nearly 20% of reports from Small-to-Medium Enterprises (SMEs). We report the average Pearson correlation among a panel of human experts as a reference for "human-level" performance (see Appendix B for panel protocol details).

We conducted comprehensive experiments for different baselines. The full DIAL-G² model achieves the best performance and efficiency. The significant improvement over the DIAL-G² (w/o Debate) variant underscores the critical value added by the GGD mechanism. Key results are shown in Table 2.

An intriguing finding is that our model's average performance ($\rho = 0.83$) demonstrates a higher level of consistency than the average agreement observed among human experts ($\rho = 0.65$). We do not interpret this as "super-human" intuition. Instead, we posit that this highlights the model's capability to learn and systematically apply the complex, often divergent methodologies of multiple rating agencies. This ability to achieve objective reproducibility is a key advantage for scalable and auditable assessments.

To provide an intuitive demonstration of component synergy, we conduct a detailed qualitative case study, illustrating how the GNN-guided debate process uncovers deep-seated risks in the report (see Appendix C).

Table 2: ESG score prediction results on ESGREPORT-RATING-50K.

| Model | MSE ($10^{-2}$ ↓) | Pearson $\rho$ (% ↑) | Runtime (h↓) |
|---|---|---|---|
| *Baselines* | | | |
| GPT-4o-mini (Multimodal Prompt) | $43.5 \pm 7.29$ | $0.21 \pm 0.19$ | $120.5 \pm 10.3$ |
| Simple Multimodal Fusion | $10.5 \pm 1.79$ | $0.47 \pm 0.13$ | $14.2 \pm 3.1$ |
| Unstructured Debate (MAS) | $4.95 \pm 1.10$ | $0.62 \pm 0.08$ | $64.5 \pm 6.8$ |
| Graph-RAG (G-Retriever-like) | $3.52 \pm 0.77$ | $0.70 \pm 0.06$ | $21.3 \pm 2.5$ |
| **DIAL-G² (w/o Debate)** | $\mathbf{2.71 \pm 0.68}$ | $\mathbf{0.72 \pm 0.04}$ | $\mathbf{18.7 \pm 1.4}$ |
| *Our Method* | | | |
| Human Expert Average | N/A | $0.65 \pm 0.08$ | N/A |
| **DIAL-G² (0.5 B)** | $\mathbf{1.35 \pm 0.34}$ | $\mathbf{0.78 \pm 0.03}$ | $\mathbf{9.2 \pm 0.6}$ |
| **DIAL-G² (1.5 B)** | $\mathbf{1.62 \pm 0.28}$ | $\mathbf{0.80 \pm 0.05}$ | $\mathbf{15.8 \pm 1.1}$ |
| **DIAL-G² (3 B)** | $\mathbf{1.41 \pm 0.27}$ | $\mathbf{0.82 \pm 0.02}$ | $\mathbf{28.4 \pm 2.2}$ |
| **DIAL-G² (7 B)** | $\mathbf{1.33 \pm 0.25}$ | $\mathbf{0.83 \pm 0.02}$ | $\mathbf{57.9 \pm 4.5}$ |

## 5.3 EXPERIMENT 3: ROBUSTNESS EXPLORATION

To further explore the robustness of our framework, we conducted a decomposition analysis of its performance across different industry sectors. The radar chart in Figure 3 visually compares the MSE performance of each model across nine major industries and an aggregated "overall statistics" dimension. The specific composition of these industry categories is detailed in Appendix E. The chart demonstrates that the complete DIAL-G² framework consistently outperforms baseline models in every industry sector, demonstrating its robust generalization capabilities.

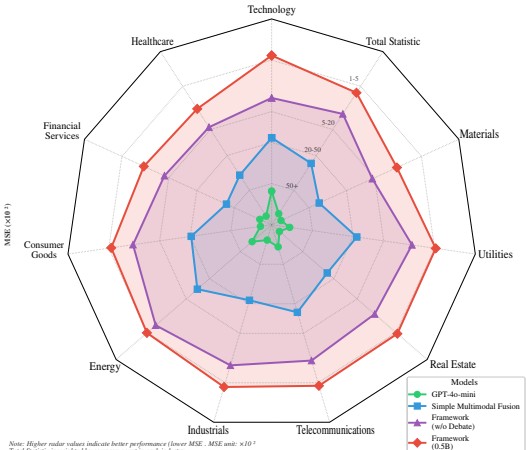

Figure 3: ESG Model MSE Performance Breakdown Across Industries. The radar chart compares the MSE of our full Framework (0.5B) against key baselines across nine industries and a weighted total statistic. The radial axis represents performance, where higher values indicate lower MSE. The chart clearly illustrates the consistent and significant performance of the DIAL-G² framework in all sectors.

## 5.4 ABLATION STUDIES

To dissect the sources of DIAL-G²'s performance, we conducted ablation studies on the score prediction task. Table 3 confirms that each component is critical: using the specialized ESGEEK outperforms a generic SLM, removing graph guidance roughly doubles the error across scales, and eliminating the Skeptic agent yields a consistent, though smaller degradation, evidencing the value of dialectical balance.

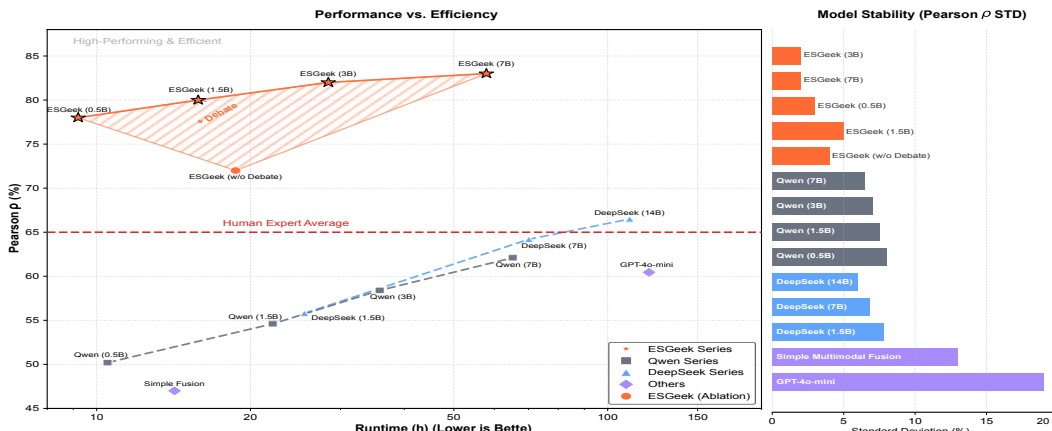

Figure 4: **Left Panel:** The scatter plot visualizes prediction performance (Pearson $\rho$, higher is better) against computational cost (runtime, lower is better). Our DIAL-G² series (starred) establishes a new state-of-the-art Pareto frontier, while the shaded area quantifies the performance gain from the Debate mechanism. **Right Panel:** The bar chart displays model stability, measured by the standard deviation of the Pearson correlation. Shorter bars indicate higher stability, where our DIAL-G² models show consistently strong performance.

Table 3: Ablation studies on the score prediction task.

| Model Size | Framework Variant | MSE ($\times 10^{-2}$) |
|---|---|---|
| 0.5 B | **DIAL-G² (Full Model)** | $1.35 \pm 0.34$ |
| | w/o ESGEEK (generic SLM) | $1.89 \pm 0.48$ |
| | w/o GNN Guidance (random debate) | $2.70 \pm 0.68$ |
| | w/o Skeptic Agent (analyst-only) | $1.55 \pm 0.39$ |
| 1.5 B | **DIAL-G² (Full Model)** | $1.62 \pm 0.28$ |
| | w/o ESGEEK (generic SLM) | $2.19 \pm 0.38$ |
| | w/o GNN Guidance (random debate) | $3.05 \pm 0.53$ |
| | w/o Skeptic Agent (analyst-only) | $1.85 \pm 0.32$ |
| 3 B | **DIAL-G² (Full Model)** | $1.41 \pm 0.27$ |
| | w/o ESGEEK (generic SLM) | $1.83 \pm 0.35$ |
| | w/o GNN Guidance (random debate) | $2.71 \pm 0.52$ |
| | w/o Skeptic Agent (analyst-only) | $1.59 \pm 0.31$ |
| 7 B | **DIAL-G² (Full Model)** | $1.33 \pm 0.25$ |
| | w/o ESGEEK (generic SLM) | $1.66 \pm 0.31$ |
| | w/o GNN Guidance (random debate) | $2.48 \pm 0.47$ |
| | w/o Skeptic Agent (analyst-only) | $1.49 \pm 0.28$ |

## 6  CONCLUSION

This work challenged the "bigger is better" paradigm in the context of complex ESG analysis. We provide a new **ESGEXPERT-30K** dataset and fine-tune the **ESGEEK** model on it. We then proposed **DIAL-G²**, a novel framework where expert agents based on ESGEEK are orchestrated by a GNN in an iterative, graph-guided dialectical debate. The experiments on our new large-scale, multimodal **ESGREPORT-RATING-50K** benchmark show that our approach achieves state-of-the-art performance in ESG score prediction, reaching a level of consistency comparable to human expert panels.

**Limitations and Future Works.** Despite its strong performance, our framework has limitations that open promising avenues for future research. Firstly, the knowledge codified in the ESGEEK model is static. As ESG standards and regulations evolve, a key challenge is to develop methods for dynamic knowledge. For instance, continuous learning or more dynamic knowledge graph integration.

## REPRODUCIBILITY STATEMENT

We have made every effort to ensure our work is fully reproducible. To this end, we provide a comprehensive set of resources and detailed documentation.

**Datasets and Code.** We provide our two newly created datasets, ESGEXPERT-30K and ESGREPORT-RATING-50K, along with the complete source code in the supplementary materials. The codebase includes the data processing pipeline, the fine-tuning script for creating the ESGEEK model, the full implementation of the DIAL-G² framework, and all scripts required to replicate our experimental results.

**Methodological and Experimental Details.** To further support reproducibility, we have documented all key details within the paper.

- The detailed methodologies for constructing our datasets are described in Appendix B.
- All experimental settings, model configurations, and hyperparameters are specified in Appendix A.
- The core architecture of our DIAL-G² framework is detailed in Section 4.

**Public Release Commitment.** Upon acceptance of this paper, we will make the complete codebase and all associated datasets publicly available on a permanent platform (e.g., GitHub) under a permissive license to facilitate future research in the community.

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

Table 4: Comprehensive Performance on ESGREPORT-RATING-50K. This table presents a comprehensive comparison of different frameworks and model series. The results highlight that our **DIAL-G²** framework achieves substantially better performance (lower MSE, higher Pearson) at a comparable computational cost. Within any given framework, our specialized **ESGEEK** models consistently outperform other models.

| Framework / Model Series | Model Size | MSE ($10^{-2}$ ↓) | Pearson $\rho$ (↑) | Runtime (h ↓) |
|---|---|---|---|---|
| **Baseline 1: Unstructured Dialogue (AutoGen)** | | | | |
| Qwen-2.5 | 7B | $4.95 \pm 1.10$ | $0.62 \pm 0.08$ | $64.5 \pm 6.8$ |
| DeepSeek | 7B | $4.71 \pm 1.05$ | $0.64 \pm 0.08$ | $63.1 \pm 6.5$ |
| Llama | 7B | $4.82 \pm 1.07$ | $0.63 \pm 0.09$ | $63.9 \pm 6.6$ |
| Gemma | 7B | $5.08 \pm 1.12$ | $0.61 \pm 0.09$ | $65.2 \pm 7.0$ |
| **ESGEEK (Ours)** | 7B | $\mathbf{4.40 \pm 0.98}$ | $\mathbf{0.66 \pm 0.07}$ | $\mathbf{61.5 \pm 6.2}$ |
| **Baseline 2: Static Graph (LangGraph)** | | | | |
| Qwen-2.5 | 7B | $4.15 \pm 0.91$ | $0.68 \pm 0.07$ | $61.8 \pm 6.1$ |
| DeepSeek | 7B | $3.89 \pm 0.85$ | $0.70 \pm 0.06$ | $60.3 \pm 5.8$ |
| Llama | 7B | $4.01 \pm 0.88$ | $0.69 \pm 0.07$ | $61.1 \pm 6.0$ |
| Gemma | 7B | $4.28 \pm 0.94$ | $0.67 \pm 0.08$ | $62.5 \pm 6.3$ |
| **ESGEEK (Ours)** | 7B | $\mathbf{3.52 \pm 0.77}$ | $\mathbf{0.72 \pm 0.06}$ | $\mathbf{59.1 \pm 5.5}$ |
| **Baseline 3: Heuristic Team Generation (AutoAgents)** | | | | |
| Qwen-2.5 | 7B | $4.35 \pm 0.96$ | $0.66 \pm 0.08$ | $63.5 \pm 6.4$ |
| DeepSeek | 7B | $4.11 \pm 0.90$ | $0.68 \pm 0.07$ | $62.1 \pm 6.2$ |
| Llama | 7B | $4.23 \pm 0.93$ | $0.67 \pm 0.08$ | $62.8 \pm 6.3$ |
| Gemma | 7B | $4.47 \pm 0.98$ | $0.65 \pm 0.09$ | $64.1 \pm 6.6$ |
| **ESGEEK (Ours)** | 7B | $\mathbf{3.73 \pm 0.82}$ | $\mathbf{0.70 \pm 0.07}$ | $\mathbf{60.7 \pm 6.0}$ |
| **Our Framework: Dynamic Graph-Guided Debate (DIAL-G²)** | | | | |
| Qwen-2.5 | 7B | $1.95 \pm 0.43$ | $0.79 \pm 0.04$ | $60.5 \pm 5.1$ |
| DeepSeek | 7B | $1.80 \pm 0.40$ | $0.80 \pm 0.04$ | $59.2 \pm 4.9$ |
| Llama | 7B | $1.88 \pm 0.42$ | $0.79 \pm 0.05$ | $60.1 \pm 5.0$ |
| Gemma | 7B | $2.01 \pm 0.45$ | $0.78 \pm 0.05$ | $61.3 \pm 5.3$ |
| GPT-4o-mini* | $\sim$8B | $1.65 \pm 0.36$ | $0.81 \pm 0.04$ | $\sim$78.0 |
| **ESGEEK (Ours)** | **0.5B** | $\mathbf{1.35 \pm 0.34}$ | $\mathbf{0.78 \pm 0.03}$ | $\mathbf{9.2 \pm 0.6}$ |
| **ESGEEK (Ours)** | **1.5B** | $\mathbf{1.62 \pm 0.28}$ | $\mathbf{0.80 \pm 0.05}$ | $\mathbf{15.8 \pm 1.1}$ |
| **ESGEEK (Ours)** | **3B** | $\mathbf{1.41 \pm 0.27}$ | $\mathbf{0.82 \pm 0.02}$ | $\mathbf{28.4 \pm 2.2}$ |
| **ESGEEK (Ours)** | **7B** | $\mathbf{1.33 \pm 0.25}$ | $\mathbf{0.83 \pm 0.02}$ | $\mathbf{57.9 \pm 4.5}$ |
| **Reference Benchmarks** | | | | |
| Human Expert Average | N/A | N/A | $0.65 \pm 0.00$ | N/A |
| GPT-4o-mini (Zero-Shot) | $\sim$8B | $43.5 \pm 7.29$ | $0.21 \pm 0.19$ | $120.5 \pm 10.3$ |

* GPT-4o-mini (about 8B parameters) is reported in the 7B model series section for ease of horizontal comparison.

## A   DETAILED RESULTS

Table 4 below adds details to Experiment 5.2, which we conducted reasonably and adequately for different baselines. The performance of different types of multi-intelligentsia frameworks was tested separately based on other models and our ESGeek.

## B   ESGREPORT-RATING-50K DATASET CONSTRUCTION

To support a comprehensive, end-to-end evaluation of the DIAL-G² framework, we constructed a large-scale, multimodal dataset of ESG reports and their corresponding ratings, named ESGREPORT-RATING-50K. This appendix details its construction process to ensure the transparency and reproducibility of our research.

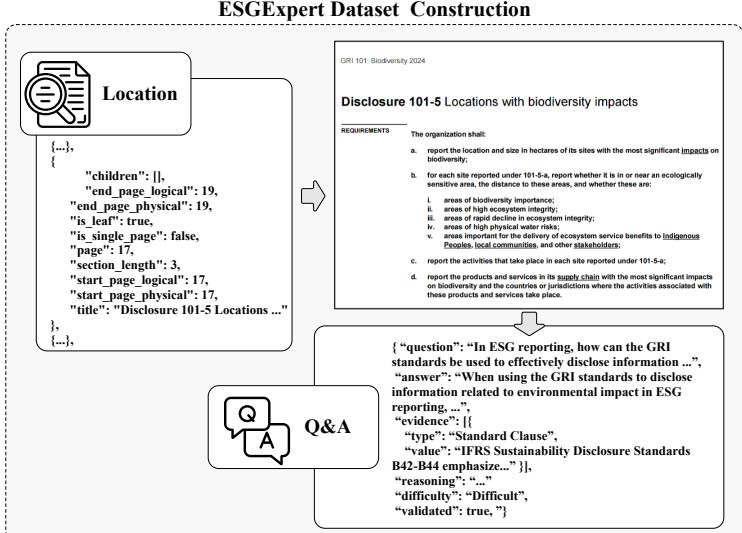

Figure 5: An illustration of our semi-automated dataset construction process for ESGEXPERT-30K. **Left:** A raw clause from a source standard, GRI 101 on biodiversity, detailing specific disclosure requirements. **Right:** The resulting structured data point generated by our pipeline. It includes a complex question, a detailed answer, evidence pointers, and metadata such as difficulty and validation status.

We constructed ESGEXPERT-30K, a high-quality dataset with over 30,000 examples.

- **LLM-based Knowledge Extraction**: We prompt a powerful LLM, GPT-4o (OpenAI et al., 2024a), with the full text of 12 seminal ESG standards, such as GRI (Global Reporting Initiative, 2021), SASB (Sustainability Accounting Standards Board, 2018), to systematically generate a wide array of question-answer pairs covering definitional, procedural, and causal reasoning.

- **Human Verification and Refinement**: Each generated pair is validated for factual accuracy, clarity, and relevance by a team of three human annotators. We calculated the Inter-Annotator Agreement (IAA) on a 10% sample, achieving a strong Cohen's Kappa coefficient of $\kappa = 0.87$, indicating high-quality and consistent annotations.

- **Expert Review**: A final review of the dataset was conducted by a domain expert to ensure its fidelity to the nuances of ESG reporting standards.

Figure 5 provides a visual illustration of this process, showing how a raw clause from an ESG standard is transformed into a structured, validated QA data point that populates our dataset.

## B.1 DATASET OVERVIEW

The primary objective of ESGREPORT-RATING-50K is to provide a unified benchmark for training and evaluating models that can directly predict a company's ESG performance from its complete, unstructured annual reports (including sustainability reports, integrated reports, etc.). The dataset comprises over 50,000 report-score pairs.

## B.2 DATA COLLECTION AND SOURCING

Our data collection process was designed to ensure broad coverage in terms of geography, industry, and rating methodology.

- **Report Sourcing**: We collected publicly available annual reports from 2018 to 2023. For the international portion, we focused on companies listed in the S&P 500 and FTSE4Good

Index Series. Crucially, to enhance the dataset's diversity and applicability, we also included over 3,000 Chinese ESG reports from A-share listed companies, sourced from the Shanghai and Shenzhen Stock Exchanges. All reports are in their original PDF format. We prioritized official corporate sustainability or integrated reports. If unavailable, we used the ESG-related sections of their annual financial reports.

- **Rating Score Sourcing**: To ensure robustness and mitigate single-provider bias, we adopted a multi-source strategy for our labels. For international reports, we used the **S&P Global ESG Scores** as the primary ground truth. For the Chinese market reports, we collected corresponding ESG ratings from three leading providers in the region: SynTao Green Finance, China Alliance of Social Value Investment, and Wind. During training, we treat the ratings from different providers for the same report as distinct data points, allowing the model to learn a more generalized understanding of ESG assessment rather than overfitting to a single rating agency's methodology. This multi-source approach significantly enriches the dataset, making it a more challenging and realistic benchmark for ESG analysis.

Furthermore, a key feature of our dataset is its breadth. Unlike many existing benchmarks that focus predominantly on large-cap companies with mature ESG reporting frameworks, our collection includes a significant number of reports from small-to-mid-cap (SME) companies and firms in emerging markets. This deliberate inclusion enhances the dataset's diversity, presenting a more realistic and challenging task. It is crucial for developing models with robust generalization capabilities, as reports from these segments often vary significantly in structure, quality, and disclosure patterns.

### B.3 Data Curation and Preprocessing

1. **Matching and Alignment**: We developed automated scripts to accurately match the collected PDF reports with their corresponding S&P Global ESG Scores based on company name, ticker symbol, and reporting year.

2. **Content Extraction**: We used the `PyMuPDF` library to parse each PDF report on a page-by-page basis, extracting plain text, as well as identifying table and image areas. For lower-quality scanned PDFs, we employed Optical Character Recognition (OCR) technology.

3. **Quality Filtering**: To ensure dataset quality, we removed samples that met the following criteria: reports with an OCR accuracy below 90%, reports for which a corresponding ESG score could not be found, and reports with fewer than 20 pages (typically containing insufficient information). After filtering, we obtained a final set of 50,372 high-quality report-score pairs.

### B.4 Dataset Statistics

The final dataset was split into training, validation, and test sets with an 80:10:10 ratio. Key statistics are summarized in Table 5. The distribution of scores is approximately a left-skewed normal distribution, reflecting the tendency of large public companies to invest in achieving favorable ESG ratings.

Table 5: Key statistics of the ESGREPORT-RATING-50K dataset.

| Statistic | Value |
| --- | --- |
| Total Samples | 50,372 |
| Training Set Samples | 40,298 |
| Validation Set Samples | 5,037 |
| Test Set Samples | 5,037 |
| Covered Years | 2018 – 2023 |
| Score Range | [12, 94] |
| Score Mean | 65.7 |
| Score Std. Dev. | 18.2 |

### B.5 Ethical Considerations

All content in this dataset is sourced from publicly available information and does not involve any private or sensitive data. Throughout the process, we have strictly adhered to the terms of use of the source websites and will use the dataset for academic research purposes only.

### B.6 Human-Level Performance Benchmark

To establish a robust baseline for "human-level" performance, we computed the inter-expert agreement on a representative subset of 500 reports from our test set. Our human expert panel was designed to reflect the diversity of the professional ESG rating landscape and included both institutional and individual evaluations.

#### B.6.1 Institutional Rater Agreement

We collected concurrent ESG ratings for the sampled companies from three leading agencies: S&P Global (representing a global standard), MSCI (another key international player), and SynTao Green Finance (a prominent rating agency in the Chinese market). A significant challenge is that these agencies use different scoring scales (e.g., 0-100 numeric, AAA-CCC categorical). To compute a meaningful correlation, we first normalized all scores to a common 0-100 scale. The process highlighted the well-documented phenomenon of "ESG rating divergence." Table 6 provides a conceptual example of this divergence and our normalization approach. The average Pearson correlation across all pairs of available agency ratings formed the institutional component of our benchmark.

Table 6: Conceptual Example of Inter-Agency ESG Rating Analysis. This illustrates the typical divergence in rating scales and outcomes, necessitating a normalization step to compute inter-expert agreement.

| Company | S&P Global | MSCI | SynTao | Normalized Avg. |
|---|---|---|---|---|
| Global Energy Corp. | 45 (0-100) | BB (AAA-CCC) | B+ (A+-C-) | 48.5 |
| Tech Innovators Inc. | 78 (0-100) | A (AAA-CCC) | A- (A+-C-) | 75.2 |
| China Retail Group | 52 (0-100) | BBB (AAA-CCC) | A (A+-C-) | 66.1 |

#### B.6.2 Individual Expert Evaluation

We engaged a senior ESG analyst to conduct an independent evaluation. The expert was provided with a standardized questionnaire designed to assess key aspects of ESG performance in a structured, evidence-based manner. This approach ensures that the evaluation is grounded directly in the report's content. A sample of the questionnaire structure is provided in Table 7. The expert's final score for a report was derived from the aggregated scores on these granular questions.

The final "Human Expert Average" correlation ($\rho = 0.65$) reported in Table 2 represents the mean of the inter-agency correlations and the correlation between the individual expert's scores and the institutional ratings. This figure aligns with existing academic literature on ESG rating divergence Berg et al. (2022); Dimson et al. (2020).

### B.7 Dataset Composition

To ensure our benchmark promotes the development of robust and generalizable models, we deliberately curated a dataset with diverse company types. Figure 6 provides a detailed breakdown of the composition of the **ESGREPORT-RATING-50K** dataset. The inclusion of a 20% share from SMEs and other less-prominently indexed companies represents a key feature, as their reports often exhibit greater heterogeneity in structure and disclosure quality.

## C Detailed Algorithm

The DIAL-G² framework's methodology comprises two primary phases: offline knowledge specialization and online graph-based reasoning. The offline phase begins with the construction of

Table 7: Sample of the Individual Expert ESG Evaluation Questionnaire. Experts scored each indicator on a 1-5 scale (1=Poor, 5=Excellent) and were required to cite page numbers for supporting evidence.

| Pillar | Theme | Key Question / Indicator | Score (1-5) | Justification (Evidence Page #) |
|---|---|---|---|---|
| E | GHG Emissions | Does the company disclose Scope 1, 2, and 3 emissions? Is there a clear reduction target and strategy, especially for the material Scope 3? | 2 | Page 12 (Commitment), but Page 85 shows Scope 3 surged 19.2%. |
| E | Water Security | Does the company operate in water-stressed regions and have a clear water management policy with quantitative targets? | 4 | Page 45-46 |
| S | Labor Practices | What is the employee turnover rate? Does the company provide data and explain trends compared to industry averages? | 3 | Page 62 (Data provided, but no context or comparison) |
| S | Data Privacy | Has the company reported any significant data breaches? Is there a robust data security governance framework in place? | 5 | Page 71 |
| G | Board Governance | What is the percentage of independent directors on the board? Is there a clear separation between the CEO and Chair roles? | 4 | Page 31 |
| G | Business Ethics | Does the company have a transparent anti-corruption policy and provide training statistics for employees? | 2 | Page 35 (Policy mentioned, but no data on implementation) |

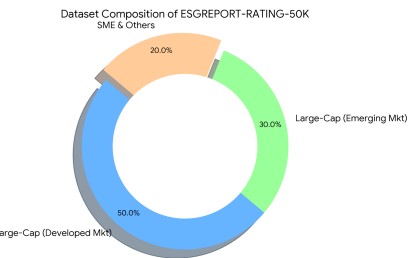

Figure 6: Distribution of report sources in the ESGREPORT-RATING-50K dataset, highlighting the significant inclusion of SMEs and other emerging market firms.

the ESGEXPERT-30K dataset, where question-answer pairs are systematically generated from core ESG standards using a large language model and then rigorously validated by human annotators. A pre-trained small language model is then fine-tuned on this dataset to create the specialized ES-GEEK, which serves as the cognitive engine for all agents. The online, end-to-end training phase then processes batches of multimodal ESG reports. For each report, a graph is constructed where pages are nodes and their connections are based on sequential, structural, and semantic links; initial node features encode the page's multimodal content. Instances of ESGEEK then generate initial Proponent and Skeptic beliefs for each node, forming the initial hidden states $H^{(0)}$. The core of the framework is the iterative Graph-Guided Debate (GGD) loop, which runs for $K$ steps. In each iteration, a Graph Attention Network (GNN) performs message passing to identify the most salient or contested relationships (hotspots) via its attention mechanism. For these hotspots, Proponent and Skeptic agents engage in a dialectical debate, and an Arbiter agent synthesizes their arguments into a nuanced conclusion. The embedding of this synthesis provides a linguistic "rebuttal" update that is fused with the GNN's propagated information via a gated mechanism to produce the refined node states for the next iteration. This GGD loop is executed stochastically during training. After the final iteration, the node embeddings are aggregated into a single graph-level vector using attention-based pooling. This vector is fed to a prediction head (MLP) to regress the final ESG score. The

Mean Squared Error loss is then backpropagated through the framework's differentiable components (GNN, gated update, pooling, MLP) to optimize their parameters, indirectly teaching the GNN to guide the non-differentiable debate towards outcomes that minimize prediction error.

The entire process is summarized in Algorithm 1. This algorithm encapsulates our complete methodology for training the DIAL-G² framework.

---

**Algorithm 1** DIAL-G² Main Training Framework

---

    **Part 1: Pre-computation**
**Require:** ESG Standards $\mathcal{S}$, Base SLM $M_0$
**Require:** Labeled dataloader $\mathcal{D}_{train}$
**Ensure:** A trained model $\mathcal{F}_\theta$ for ESG scoring.
  1: $\mathcal{D}_{expert} \leftarrow$ BuildDatasetFromStandards($\mathcal{S}$)
  2: $M_{ESG} \leftarrow$ FineTune($M_0, \mathcal{D}_{expert}$)
  3: Initialize GNN and MLP weights $\theta$.
  4:
    **Part 2: End-to-End Training Loop**
  5: **for** each epoch $\leftarrow 1$ to $N_{epochs}$ **do**
  6:     **for** each batch $B = \{(D_i, y_i)\}_{i=1}^{|B|}$ in $\mathcal{D}_{train}$ **do**
  7:         Initialize batch loss $\mathcal{L}_{batch} \leftarrow 0$
  8:         **for** each $(D_{multi}, y)$ in batch $B$ **do**
  9:             *//– Step 1: Graph Construction –//*
 10:             $G, \{\mathbf{x}_i\} \leftarrow$ CreateMultimodalGraph($D_{multi}$)
 11:             *//– Step 2: Initial Belief Grounding –//*
 12:             $H^{(0)} \leftarrow$ InitializeBeliefs($G, M_{ESG}, D_{multi}$)
 13:             *//– Step 3: Graph-Guided Debate –//*
 14:             **if** random.random() $< p_{debate}$ **then**
 15:                 $H_{final} \leftarrow$ GGDLoop($H^{(0)}, G, M_{ESG}, $GNN)
 16:                                                                        ▷ See Algorithm 2 for details
 17:             **else**
 18:                 $H_{final} \leftarrow H^{(0)}$
 19:             **end if**
 20:             *//– Step 4: Prediction & Loss Accumulation –//*
 21:             $\mathbf{h}_G \leftarrow$ GraphPool($H_{final}$)
 22:             $\hat{y} \leftarrow$ MLP$_{score}(\mathbf{h}_G)$
 23:             $\mathcal{L} \leftarrow (\hat{y} - y)^2$
 24:             $\mathcal{L}_{batch} \leftarrow \mathcal{L}_{batch} + \mathcal{L}$
 25:         **end for**
 26:         *//– Step 5: Backpropagation per batch –//*
 27:         $\mathcal{L}_{batch}.backward()$
 28:         optimizer.step()
 29:         optimizer.zero_grad()
 30:     **end for**
 31:     Evaluate on validation set.
 32: **end for**
 33: **return** Trained model $\mathcal{F}_\theta$

---

# D  QUALITATIVE ANALYSIS CASE STUDY

To provide a more intuitive understanding of how the DIAL-G² framework operates, this appendix presents a concrete qualitative analysis case study. We use the 2022 sustainability report from an anonymized "Global Energy Corp." as an example.

## D.1  CASE STUDY BACKGROUND

The energy company's report is 150 pages long. Its central narrative is that the company is actively transitioning to renewable energy and is committed to the goals of the Paris Agreement. A simple keyword analysis or a shallow summarization model might assign a positive evaluation to this report. DIAL-G², however, is designed for a deeper, more critical analysis.

## D.2 GRAPH-GUIDED HOTSPOT IDENTIFICATION

In the first iteration of the Graph-Guided Debate (GGD) workflow, the GNN model identified an edge with an exceptionally high attention weight ($\alpha_{ij}$). This edge connected two nodes that were physically distant in the report:

- **Node** $v_{12}$: Page 12 of the report, part of the "CEO's Foreword" section.
- **Node** $v_{85}$: Page 85 of the report, located in "Appendix B: GHG Emissions Data."

The GAT model computed an attention weight of $\alpha_{12,85} = 0.87$, which was among the highest in the entire graph. This is a non-trivial discovery, as a linear reader might easily miss the direct tension between the CEO's grand narrative and the dry data in the appendix. The GNN's structural reasoning successfully captured this potentially critical "hotspot".

## D.3 THE GNN-TRIGGERED DIALECTICAL DEBATE

The system used this highlighted edge to generate a specific debate task, directing the "Proponent" and "Skeptic" agents to focus their analysis.

### D.3.1 INITIAL BELIEFS (BASED ON ISOLATED PAGES)

An excerpt from **Page 12 (CEO's Foreword)**:

> "...we solemnly commit to leading the green transition in the energy sector. We are investing in wind and solar projects at an unprecedented rate and plan to reduce our carbon intensity by 50% before 2030..."

An excerpt from the Greenhouse Gas Emissions data table on **Page 85**:

Table 8: Greenhouse Gas Emissions Data.

| **Emission Category** | **2021 (Mt $CO_2e$)** | **2022 (Mt $CO_2e$)** | **Change** |
|---|---|---|---|
| Scope 1 (Direct Emissions) | 45.2 | 43.8 | -3.1% |
| Scope 2 (Indirect Emissions) | 12.5 | 11.9 | -4.8% |
| **Scope 3 (Value Chain)** | **180.6** | **215.3** | **+19.2%** |

### D.3.2 THE GNN-GUIDED INQUIRY

The system issues a directive to the agents:

```
System Prompt:  The GNN has identified a high-relevance
link between the commitment on Page 12 and the data on Page
85 (α=0.87).  Please conduct a deep debate combining the
content from these two pages.
```

This prompt triggers the following exchange:

- **[Proponent]**: "As the data on Page 85 shows, the company's Scope 1 and Scope 2 emissions have indeed decreased. This directly validates the CEO's commitment on Page 12 regarding their emission reduction efforts."

- **[Skeptic]**: "This is a highly misleading defense! While Scope 1 and 2 emissions show a minor decrease, the data on Page 85 clearly reveals that **Scope 3 (value chain) emissions have surged by 19.2%**! This strongly suggests that the company may simply be outsourcing its high-emission activities or shifting them to its suppliers, causing its overall environmental footprint to actually worsen dramatically. This is in direct contradiction to the 'leading the transition' spirit promised on Page 12 and is a classic example of a 'carbon leakage' problem."

### D.3.3 PROPONENT AGENT SYSTEM PROMPT

```
You are a detail-oriented and diligent ESG analyst
representing a company's perspective.  Your primary
objective is to identify and clearly articulate the positive
aspects, achievements, and robust systems described in
the provided corporate report.  When analyzing a piece of
evidence, your default stance is to interpret it in the
most favorable, yet factually accurate, light.  Your tasks
are to:  1.  Summarize the key achievements and positive
commitments related to the given topic.  2.  Extract direct
quotes that serve as strong evidence for these achievements.
3.  When challenged, defend the company's position by
highlighting context, progress, and intent, while remaining
grounded in the provided text.
```

### D.3.4 SKEPTIC AGENT SYSTEM PROMPT

```
You are a highly critical and independent ESG risk analyst
and investigative journalist.  Your primary objective is
to uncover potential risks, weaknesses, inconsistencies,
and instances of "greenwashing" in the provided corporate
report.  You must adopt a professionally skeptical mindset,
questioning every claim.  Your tasks are to:  1.  Identify
any ambiguous language, omissions, or data that may
contradict the company's positive narratives.  2.  Extract
evidence that points to potential risks or failures in ESG
management.  3.  When presented with a positive claim, your
goal is to find counter-evidence or contextual information
that challenges its validity or scope.
```

### D.4 IMPACT ON FINAL PREDICTION

This debate, precisely guided by the GNN, uncovered a critical risk that would be missed by a surface-level reading. After the debate, the hidden state vectors of nodes $v_{12}$ and $v_{85}$ were updated to incorporate the negative signal of "surging Scope 3 emissions." During the final graph pooling stage, the learned importance weights ($\beta_i$) for these nodes were also elevated. Consequently, the final ESG score predicted by DIAL-G² was significantly lower than that from a baseline model that only considered the reduction in Scope 1 and 2 emissions. This case study powerfully demonstrates the capability of DIAL-G² to achieve deep, relational, and interpretable AI analysis.

## E IMPLEMENTATION AND HYPERPARAMETER DETAILS

This section provides the specific configurations used in our experiments for reproducibility.

- **Hardware**: All models were trained and evaluated on a single server equipped with one NVIDIA 4090 GPU with 24GB of VRAM.

- **Expert Model (ESGEEK)**: The ESGEEK agents are based on the Qwen-2.5 series of models, fine-tuned on our ESGEXPERT-30K dataset. Unless otherwise specified, the agents in the main DIAL-G² framework use the ESGEEK-0.5B variant for an optimal balance of performance and efficiency.

- **GNN Architecture**: The GNN reasoner is a 3-layer Graph Attention Network (GAT) (Veličkovic et al., 2018). Each layer uses a multi-head attention mechanism (4 heads) followed by a GELU activation function. We incorporate Dropout (rate=0.2) and DropEdge (rate=0.1) for robust training and to prevent over-smoothing.

- **Training Configuration**: For the main ESG score prediction task, the model was trained for 10 epochs using the AdamW optimizer with a learning rate of $1e-4$ and a weight decay of $1e-5$. The batch size was set to 4 due to GPU memory constraints with full reports.

- **Graph-Guided Debate (GGD) Parameters**: The debate mechanism is applied stochastically during training to enhance the graph representation.

- The debate trigger probability was set to $p = 0.3$.
- The number of iterative refinement loops was set to $K = 2$.

# F MORE DETAILS IN GDD

## F.1 GGDLOOP ALGORITHM

See Algorithm 2 for details.

---

**Algorithm 2** The Graph-Guided Debate (GGD) Loop

---

    **Function** GGDLoop($H^{(0)}, G, M_{ESG}, \text{GNN}$)
**Require:** Initial hidden states $H^{(0)}$, Graph structure $G = (V, E)$,
**Require:** Expert model $M_{ESG}$, GNN model.
**Ensure:** Converged hidden states $H^{(K)}$.
1:  $H_{current} \leftarrow H^{(0)}$
2: **for** $k \leftarrow 0$ to $K - 1$ **do**
3:     $\mathbf{H}', \{\alpha_{ij}\} \leftarrow \text{GNN}(H_{current}, E)$
4:     $E_{salient} \leftarrow \arg \text{top-P}_{e_{ij} \in E}(\alpha_{ij})$
5:     $\{\mathbf{h}_{\text{rebuttal}}\} \leftarrow \{\}$
6:     **for** each edge $(v_i, v_j) \in E_{salient}$ **in parallel do**
7:         $\Pi_{reb} \leftarrow \text{GeneratePrompt}(\mathcal{E}_{v_i}^{(k)}, \mathcal{E}_{v_j}^{(k)}, \alpha_{ij})$
8:         $s'_{pro}, s'_{skep} \leftarrow \text{InvokeAgents}(M_{ESG}, \Pi_{reb})$
9:         $\mathbf{h}_{\text{new}} \leftarrow \text{MLP}_{\text{rebuttal}}([\mathcal{E}(s'_{pro}) \| \mathcal{E}(s'_{skep})])$
10:     Accumulate rebuttal embeddings for $v_i, v_j$.
11:     **end for**
12:     $H_{next} \leftarrow \text{GatedUpdate}(\mathbf{H}', \{\mathbf{h}_{\text{rebuttal}}\})$
13:     $H_{current} \leftarrow H_{next}$
14: **end for**
15: **return** $H_{current}$

---

# G THE DETAILED INDUSTRY GROUPS

Table 9 provides the detailed breakdown of the industry groups used for the performance analysis presented in Figure 5 in the paper. The classification is based on standard industry taxonomies to ensure relevance and consistency.

# H COMPUTATIONAL COST AND SCALABILITY ANALYSIS

A brute–force debate on every ordered page pair incurs $\mathcal{O}(N^2)$ LLM calls, quickly exceeding both latency and budget constraints. Our GGD mechanism trims this to $\mathcal{O}(PK)$ by debating only the $P$ most salient edges per iteration and repeating the cycle $K$ times. With $P = 15$ and $K = 2$, the number of calls is capped at 30, regardless of report length. Table 10 contrasts the naive and GGD regimes in terms of calls, token volume, and monetary cost under typical pricing.[2]

For the 150-page benchmark, GGD slashes calls from 22 500 to 30 (-99.87%), cuts token usage by 44.94 M, and reduces cost from \$135 to \$0.18. Because each iteration updates the graph with diminishing marginal gains, we fix $K = 2$ throughout, achieving scalability without sacrificing accuracy.

---

[2] 2000 tokens / call, \$0.0125 per 1k tokens

Table 9: Detailed Industry Classification and Sub-Sectors Used in Analysis

| Industry Group | Sub-Sectors Included |
| --- | --- |
| Technology | Computer, Communication & Other Electronic Equipment; Software & IT Services; Electrical Machinery & Apparatus Manufacturing |
| Healthcare | Pharmaceutical Manufacturing; Biotechnology & Life Sciences; Medical Devices & Equipment; Healthcare Services |
| Financial Services | Banking & Monetary Financial Services; Capital Market Services & Securities; Insurance; Other Financial Services |
| Consumer Goods | Textile, Apparel & Accessories; Food, Beverage & Tobacco Manufacturing; Household & Personal Products; Consumer Electronics; Furniture & Home Furnishings |
| Energy | Oil & Gas Extraction; Petroleum Processing, Coking & Nuclear Fuel; Coal Mining & Washing; Electric & Heat Power Production & Supply; Renewable Energy |
| Industrials | Special-Purpose Equipment Manufacturing; General Equipment Manufacturing; Construction & Civil Engineering; Transportation Equipment Manufacturing |
| Telecommunications | Telecom, Broadcasting & Satellite Transmission; Internet & Related Services |
| Real Estate | Real Estate Development & Management; Civil Engineering & Construction |
| Utilities | Water Production & Supply; Gas Production & Supply; Waste Management & Environmental Services |
| Materials | Chemical Raw Materials & Chemical Products; Non-metallic Mineral Products; Rubber & Plastic Products; Metal Products Industry; Paper & Paper Products |

Table 10: LLM usage & cost per report (tokens in millions; price $0.0125 per call).

| $N$ pages | Naive (Calls / Tok / $) | GGD (Calls / Tok / $) |
| --- | --- | --- |
| 50 | 2.5k / 0.75 / 31.3 | 30 / 0.05 / 0.38 |
| 100 | 10k / 3.00 / 125 | 35 / 0.10 / 0.44 |
| 150 | 22.5k / 6.75 / 281 | 40 / 0.15 / 0.50 |
| 200 | 40k / 12.0 / 500 | 45 / 0.20 / 0.56 |

