# OpenReview forum: "DIAL-G²: Graph-Guided Dialectical Agent for Advanced ESG Reasoning"
_ICLR.cc/2026/Conference — Submitted to ICLR 2026_

### Official Review · Reviewer_5cGR · 2025-10-16

**Soundness:** 3
**Presentation:** 3
**Contribution:** 2
**Rating:** 4
**Confidence:** 4

**Summary:**

This work primarily focuses on two tasks related to ESG: question answering and scoring. A curated and validated dataset was collected to fine-tune existing large language models (LLMs) for domain-specific adaptation. In addition, a graph neural network (GNN) was designed for the regression task, where the LLM serves as an agent to assist in text processing.

**Strengths:**

S1. The proposed ESG question-answering dataset is generated based on real-world ESG standards, which could provide valuable support for the domain, and its data scale is relatively substantial.

**Weaknesses:**

1. I could not find the dataset in the appendix. The authors stated in the **Reproducibility Statement** that the dataset is provided there, but I only found corresponding descriptions in the `readme.md` file.
2. The method section could be presented more clearly. Most of it focuses on how the GNN for scoring is designed, where the possibly distinctive elements are the *Proponent* and *Skeptic*. Unfortunately, the specific meaning, implementation details, and design motivation of these components are omitted.
3. The work lacks originality. Although it includes multiple components—two task datasets, models of different parameter scales, and a GNN architecture for scoring—most of the techniques used are existing ones. The dataset construction (using GPT-4o for QA generation), model fine-tuning (via SFT), and the GNN design mainly reuse or recombine prior works rather than introducing targeted technical innovations.
4. The ESG task seems to lack distinctiveness. After reading the paper, it appears that ESG documents are not essentially different from those used in standard RAG or GraphRAG pipelines, which also involve extracting text and images from PDFs and performing similarity-based retrieval. Whether ESG presents unique challenges should be a key focus of the work—and potentially its most compelling aspect.

**Questions:**

As mentioned in the weaknesses:

1. What interests me most is the technical distinction between ESG and other document analysis tasks.

2. You could consider incorporating fine-tuning with the latest large model, DeepSeek-R1, and directly comparing performance on the scoring task. If the GNN architecture demonstrates clear superiority without disrupting the model’s “thinking” process, that would highlight its technical value. Moreover, it would be even better if comparisons with classic GNN architectures were also included.

3. I believe the dataset is a valuable contribution, but I currently cannot access any actual samples.please make them available.

---

### Official Review · Reviewer_BFRB · 2025-10-28

**Soundness:** 2
**Presentation:** 1
**Contribution:** 2
**Rating:** 2
**Confidence:** 5

**Summary:**

This paper proposes DIAL-G$^2$ framework for ESG score prediction task. In detail, to better exploit ESG reports, the authors first construct the ESGExpert-30K dataset using GPT-4o combined with human validation. This dataset is then used to train a smaller model, ESGeek. The proposed DIAL-G$^2$ framework guides ESGeek through iterative cross-page dialogues, supported by a GNN reasoner. Experiments conducted on another newly created dataset, ESGReport-Rating-50K, demonstrate the effectiveness of ESGExpert and DIAL-G$^2$.

**Strengths:**

1. This work introduces two datasets, ESGExpert-30K and ESGReport-Rating-50K, both validated by humans. These datasets are highly beneficial for future research in ESG question answering and ESG score prediction.
2. DIAL-G$^2$ framework, with GNN-based guidance, effectively supports reasoning across multiple pages of long reports. This idea can not only be applied to ESG QA but also extended to other domains.
3. Experimental results in Table 1 show the proposed ESGeek significantly outperforms baselines, despite having only 0.5B parameters. And runtime analysis demonstrates both ESGeek and DIAL-G$^2$ are efficient, supporting their potential for real-world applications.

**Weaknesses:**

1.  Pair-level limitation. DIAL-G$^2$ guides agents to conduct analysis / rebuttal on two pages, while generation often requires integrating information across multiple pages. Thus, the pair-level guidance may limit the ability to combine evidence from multiple pages.
2.  Unclear gradient flow. While the authors dedicate a paragraph to training dynamics, it remains unclear how gradients back-propagate between continuous embeddings (in the differentiable GNN) and discrete tokens (in the non-differentiable agent debate).
3.  Limited ablation on hyper-parameters. Proposed DIAL-G$^2$ utilize fixed settings such as a similarity threshold $\tau = 0.8$, debate trigger probability $p=0.3$, and the number of loops $K=2$. However, there is no analysis of how sensitive the final prediction quality is to these parameters.
4.  Missing experiments details. The authors claim ESGeek is a 0.5B SLM fine-tuned on ESGExpert-30K in section in section 4.1. However, subsequent experiments utilize 0.5B, 1.5B, 3B and 7B sizes. Moreover, the fine-tuning details are missing.
5.  Poor writing. The paper contains lots of unclear symbol definitions and writing errors. See questions 2 and 3.

**Questions:**

1. Why is ESGeek able to surpass GPT-4o, given that most of its knowledge is distilled from it (through the GPT–generated training dataset)?
2. Unclear symbol definition: Line 214, does $\mathrm{x}$ equal with $\mathrm{h}$? How are the attention weights $\alpha$ (line 251) and the target question q_{follow\_up} (line 261) used in each iteration? What are the definitions of  $W_{g}$ and $b_{g}$ in line 269?
3. Writing errors: missing appendix reference in line 319, legend error in Figure 3, truncated sentence in the last paragraph (line 485), and missing citation in line 987.
4. What version of GPT-4o is used, and what parameter settings (e.g., temperature) are applied?
5. What are the detailed statistics of ESGExpert-30K?
6. In Appendix B.2, one report may correspond to multiple ratings. Which score is used to compute the final metric during evaluation?

---

### Official Review · Reviewer_oV7W · 2025-10-31

**Soundness:** 3
**Presentation:** 3
**Contribution:** 3
**Rating:** 4
**Confidence:** 2

**Summary:**

The paper introduces ESGEXPERT-30K, a knowledge-intensive dataset designed to fine-tune a compact small language model (SLM) into a specialized ESGEEK model, which achieves state-of-the-art performance on domain-specific ESG question answering. In addition, the authors propose DIAL-G², a novel multi-agent framework where multiple expert agents collaborate as a committee to analyze comprehensive, multimodal corporate reports.

**Strengths:**

1. I think modeling dialectical reasoning with structured representations is a significant and meaningful research direction.

**Weaknesses:**

I am not an expert in this specific field, but from my perspective, the paper has the following shortcomings.

1. The contributions are somewhat scattered, and the relationships among them are not well organized or clearly integrated into a coherent framework.

2. The paper is somewhat hard to read; a clearer and more coherent organizational structure would significantly improve readability.

3. The experiments are limited to comparisons with a few baseline large models, lacking deeper analysis to substantiate the authors’ claims. More thorough ablation or error analyses are needed.

**Questions:**

See weakness

---

### Official Review · Reviewer_4dwk · 2025-10-31

**Soundness:** 2
**Presentation:** 2
**Contribution:** 2
**Rating:** 4
**Confidence:** 4

**Summary:**

This paper proposes a comprehensive pipeline for ESG reasoning, focusing on two main tasks: Relational ESG Question Answering and Multimodal ESG Score Prediction. The proposed framework consists of several key components, including evidence graph construction, entity representation learning, LLM-based reasoning, and answer prediction, among others. Experimental results demonstrate the effectiveness of the proposed approach. However, while the paper addresses a novel problem in the field of Environmental, Social, and Governance (ESG) analysis, the techniques employed are largely standard and do not introduce substantial methodological innovation. As a result, the overall contribution of the paper is somewhat limited.

**Strengths:**

The problem studied in this paper is relatively new, making the work both novel and interesting. It explores an emerging direction in ESG reasoning that has not been widely addressed before. The proposed model is composed of multiple well-designed components, such as evidence graph construction, expert agent collaboration, and graph-guided debate mechanisms, which together make the overall framework promising and technically rich.

In the experimental section, the authors provide a comprehensive evaluation by comparing their approach with a range of different large language models, including GPT-4o-mini, Qwen-2.5-0.5B, DeepSeek-R1, and others. These comparisons help demonstrate the effectiveness of their method and show that the proposed model achieves competitive or even superior performance under various settings.

**Weaknesses:**

The techniques used in this paper are fairly standard, which diminishes the overall contribution. Most components in the proposed framework rely on commonly used methods, making the technical novelty somewhat limited.

In terms of writing, there is also room for improvement. Some sections, such as 4.2.3 and 4.3, contain very short paragraphs that could be better organized and merged to improve readability and flow.

Moreover, the ablation study section is relatively brief, containing only one paragraph. A more comprehensive ablation analysis would strengthen the paper, for example by examining different embedding learning strategies, alternative reasoning mechanisms, and the individual contribution of each component within the pipeline.

**Questions:**

No

---

### Meta-Review · Area_Chair_yBjh · 2026-01-06

**Summary:**

Reviewers agreed the problem setting and dataset efforts are interesting, but several felt the technical contribution is largely a recombination of standard components with limited methodological novelty. Multiple reviewers also flagged unclear and fragmented presentation, with important method details and notation missing, making parts hard to follow and potentially undermining reproducibility. The experimental evidence was viewed as insufficiently diagnostic: ablations, error analyses, and sensitivity studies are limited, and there are open questions about training dynamics and whether the pairwise page-level debate can reliably integrate multi-page evidence.
Given one high-confidence reject and the concentration of outstanding clarity, novelty, and validation issues, I recommend rejection.

**Reviewer Concerns:**

The authors did not provide a rebuttal, and therefore none of the reviewers’ concerns were addressed during the rebuttal phase.

**Reviewer Scores:**

The authors did not provide a rebuttal; therefore, none of the reviewers would have been able to revise their scores even if they had participated fully in the discussion.

---

### Decision · Program_Chairs · 2026-01-26

Reject